# Atomic Force Microscopy Investigation of the Interactions between the MCM Helicase and DNA

**DOI:** 10.3390/ma14030687

**Published:** 2021-02-02

**Authors:** Amna Abdalla Mohammed Khalid, Pietro Parisse, Barbara Medagli, Silvia Onesti, Loredana Casalis

**Affiliations:** 1Elettra-Sincrotrone Trieste, 34149 Trieste, Italy; amnaphysics@gmail.com (A.A.M.K.); bmedagli@units.it (B.M.); 2Department of Physics, PhD School in Nanotechnology, University of Trieste, 34127 Trieste, Italy; 3Istituto Officina dei Materiali, Consiglio Nazionale delle Ricerche (IOM-CNR), 34149 Trieste, Italy; 4Department of Chemical and Pharmaceutical Sciences, University of Trieste, 34127 Trieste, Italy

**Keywords:** DNA–protein interaction, MCM helicase, AFM single molecule imaging, biopolymers mechanics

## Abstract

The MCM (minichromosome maintenance) protein complex forms an hexameric ring and has a key role in the replication machinery of Eukaryotes and Archaea, where it functions as the replicative helicase opening up the DNA double helix ahead of the polymerases. Here, we present a study of the interaction between DNA and the archaeal MCM complex from *Methanothermobacter thermautotrophicus* by means of atomic force microscopy (AFM) single molecule imaging. We first optimized the protocol (surface treatment and buffer conditions) to obtain AFM images of surface-equilibrated DNA molecules before and after the interaction with the protein complex. We discriminated between two modes of interaction, one in which the protein induces a sharp bend in the DNA, and one where there is no bending. We found that the presence of the MCM complex also affects the DNA contour length. A possible interpretation of the observed behavior is that in one case the hexameric ring encircles the dsDNA, while in the other the nucleic acid wraps on the outside of the ring, undergoing a change of direction. We confirmed this topographical assignment by testing two mutants, one affecting the N-terminal β-hairpins projecting towards the central channel, and thus preventing DNA loading, the other lacking an external subdomain and thus preventing wrapping. The statistical analysis of the distribution of the protein complexes between the two modes, together with the dissection of the changes of DNA contour length and binding angle upon interaction, for the wild type and the two mutants, is consistent with the hypothesis. We discuss the results in view of the various modes of nucleic acid interactions that have been proposed for both archaeal and eukaryotic MCM complexes.

## 1. Introduction

Understanding at the molecular level the mechanisms that govern DNA replication in proliferating cells is fundamental to understand diseases connected to genomic instabilities, such as genetic syndromes and cancer. A key step for DNA replication to take place is the separation of the two antiparallel strands; this is catalyzed by an important class of enzymes called helicases. Helicases unwind the DNA duplex as the replication fork moves, using energy from ATP hydrolysis to drive the process, affecting the overall topology of the DNA, with important implications especially in relation to operations in real, crowded systems. Biochemical analyses are often not sufficient for topological detailed characterizations, which also requires biophysical and structural tools. Among those, atomic force microscopy (AFM) has proven very useful to visualize and characterize the interaction between proteins and nucleic acids in the physiological environment, through the careful statistical analysis of single DNA molecules contour length and bend angle variations upon interactions [1,2,3,4,5,6].

We chose here to focus on the archaeal MCM helicase, which has a critical role in the initiation and progression of DNA replication and thus interacts with DNA in a dynamic and challenging pattern during the various stages of the duplication process. The MCM (minichromosome maintenance) proteins belong to the AAA+ family and are conserved from Archaea to higher eukaryotes. In eukaryotic cells, six of these proteins (called MCM2, MCM3, MCM4, MCM5, MCM6, and MCM7) assemble into a hexamer acting as the replicative helicase [7,8,9]. Archaea present a simplified version of the eukaryotic DNA replication system; in most Archaea, the helicase is a single MCM protein, forming a homohexamer (Figure 1) [10,11,12,13,14]. The archaeal complex shows a processive ATP-dependent helicase activity in the 3′→5′ direction. Whereas the archaeal complex has an intrinsic helicase activity, the eukaryotic MCM2-7 complex requires two additional components: the Cdc45 replication factor and the GINS tetrameric complex [15,16,17].

A canonical MCM protein can be divided into three domains (Figure 1a): N-terminal, AAA+, and C-terminal domains. The N-terminal domain is composed by three subdomains: subdomain A (sA) is the external part of the ring and is involved in the regulation of the helicase activity, subdomain B contains a structural Zn motif, and subdomain C is responsible for the hexamerization and contains a positively-charged β-hairpin (NBH, Figure 1c,d) extending towards the centre of the channel, involved in DNA binding [18,19]. The AAA+ domain is the catalytic core of the enzyme and is responsible for ATP hydrolysis and DNA unwinding, whereas a C-terminal domain folds into a winged helix domain and may assist in DNA binding. The hexameric ring shows two tiers, one corresponding to the N-terminal domain, and the other to the catalytic AAA+ domain, with the nucleotide binding at the interface between the monomers (Figure 1b). The DNA threads through the central channel of the ring (Figure 1c).

The archaeal MCM complex can assume a large number of conformations, as visualized by electron microscopy and crystallography, including hexamers, heptamers, double hexamer/heptamer structures, and a variety of open rings and/or helical assemblies [10,11,12,20,21,22,23,24]. Over the last decade a wealth of results has contributed to the understanding of the intricate mechanism of action of MCM proteins, alone and within the CMG assembly [9,15,16,25]. Nevertheless, the complexity of this essential machinery is such that many questions are still open: among those are the multiple modes of interaction between the MCM hexamer with DNA, at different stages along the reaction pathway. As in all the hexameric helicases [26] the DNA threads into the central channel as the complex moves along and unwinds (Figure 1c); however alternative modes of interaction with DNA have been suggested, based on biochemical data, macromolecular crystallography and electron microscopy [27,28]. Atomic force microscopy is a powerful method to distinguish upon the different mechanisms of interaction; although not providing atomic resolution in air/physiological environment, it has the advantage of showing a direct image of the interaction, and does not require the averaging steps that are necessary for instance in single particle cryo-electron microscopy. AFM can therefore provide complementary information at the single-molecule level, and offers biologists and biophysicists an additional tool to build a multifaced picture of a biological process.

Here we carried out biophysical studies using single molecule AFM imaging, to visualize the protein–DNA interaction as a function of different interaction parameters. After a careful optimization of the surface immobilization strategy of DNA and MCM complexes, we investigated the DNA–MCM interaction using AFM topography imaging in air, highlighting different modes of interaction. The use of two mutant forms of the protein provides some clue as to the possible significance of these binding modes. From the statistical analysis of the bending angles and the end-to-end distance of DNA molecules in the free status, or in interaction with the MCM complex and its mutants, we obtained information about the conformational changes induced by the presence of the protein complex, and discussed the results in view of the current literature.

## 2. Materials and Methods

### 2.1. Expression and Purification Protocol of Wild-Type MthMCM and Mutants

An expression vector with the *Methanothermobacter thermautotrophicus* MCM (MthMCM) gene cloned into the pET21b expression vector, to express a C-terminal His-tagged protein with an uncleavable tag, was a gift by J. Gautier, Columbia University. The vectors to express the mutant proteins lacking subdomain A (ΔsA, missing the first 91 amino-acid residues) and the NBH mutants (where the two positively-charged residues R227 and K229, belonging to the N-terminal β-hairpin, were replaced by Ala) were kindly provided by the laboratory of Z. Kelman (University of Maryland, Rockville, MD, USA). All the proteins were expressed in *E. coli* BL21 cells and expressed and purified as previously described [14,27].

### 2.2. DNA Substrate Preparation

DNA substrates of various length (250 bp, 495 bp, 807 bp, 1000 bp, 1200 bp and 1350 bp) were obtained by polymerase chain reaction (PCR) using a touch down method to avoid amplifying nonspecific sequences, monitored using agarose gel electrophoresis and purified with a PCR purification kit (Qiagen, Venlo, the Netherlands). Protein and DNA concentrations were measured using a NanoDrop 2000c UV–vis spectrophotometer (Thermo Scientific, Waltan, MA, USA).

### 2.3. Sample Preparation

We used freshly cleaved mica, an atomically flat surface (roughness: 0.1 nm), which is commonly used as a substrate for AFM studies of DNA and proteins. Since DNA and mica are both negatively charged in buffer solution, their interaction needs to be mediated by opposite charges, either absorbed on the surface (e.g., NH^+2^ groups) or present in solution (divalent cations) [1,2]. We used freshly cleaved mica surface, on which the DNA is trapped through the divalent Mg^2+^ ions from a buffer containing MgCl_2_. DNA was diluted to a concentration of 2 nM in a buffer suitable for the protein complexes as well (9 mM MgCl_2_, 90 mM NaCl, 30 mM HEPES and 5% glycerol. pH: 6.8). Glycerol was introduced to increase protein stability for further DNA–MCM interaction studies. A 20 μL droplet of DNA solution was then incubated on freshly cleaved mica for one minute, rinsed with 5 mL MilliQ water to remove unbound DNA molecules and then the sample was dried by means of nitrogen gas. Similarly for the DNA–protein complexes interaction experiments, DNA was diluted to a concentration of 2 nM in buffer and protein complex added (in concentrations of 10 nM for the wild-type and 20 nM for the mutants, see main text). Solutions were kept at room temperature for 30 min. A 20 μL droplet of DNA–protein complex solution was then deposited onto freshly cleaved mica for one minute, rinsed with 5 mL MilliQ water to remove unbound molecules and then dried using nitrogen gas.

### 2.4. AFM Imaging and Image Processing

AFM images were acquired in air at room temperature using either an MFP3-3D (Asylum Research/Oxford Instruments, Goleta, CA, USA) or a SOLVER_Pro NT-MDT machine using either NSG30 (NT-MDT, Apeldoorn, The Netherlands, typical resonance frequency: 320 kHz; spring constant: 40 N/m), NSG03 (NT-MDT, Apeldoorn, The Netherlands), typical resonance frequency: 90 kHz; spring constant: 1.74 N/m), or AC240 (Olympus, Tokyo, Japan), resonance frequency: 70 kHz; spring constant: 2 N/m) cantilevers. All the AFM images were processed using WSxM software [29]; the contour length analysis was performed via 2D single molecule software [30] while and Igor Pro was used for the statistical analysis.

### 2.5. DNA End-to-End Distance Analysis

The analysis of DNA contour length (L) and end-to-end distance (R) was performed as described in Reference [31]. Briefly, we treated our DNA fragments as a polymer and we used the worm-like chain (WLC) model to describe the entropic elasticity of long polymer molecules. According to the WLC model [32], the mean-square end-to-end distance (<R^2^>) can be written as:(1)$$\langle R^{2}\rangle_{2D}=4PL\left(1-\frac{2P}{L}\left(1-\;e^{-\frac{L}{2P}}\right)\right)$$
where the end-to-end distance R and the DNA contour length L can be measured from AFM images. The DNA persistence length P is a basic mechanical property quantifying the stiffness of a polymer, and can be derived from Equation (1). The persistence length of free double stranded DNA in a high salt solution is known to be about 45–60 nm, depending on salt conditions, temperature, and DNA length [33,34,35]. Given certain solution and surface conditions, DNA molecules can equilibrate the surface prior to binding or being trapped by the surface without equilibration (3D–2D conformation projection). If by fitting the L-R data collected from AFM images using Equation (1) we derived a persistence length value consistently close to the expected one, we concluded that the DNA did freely equilibrate on the 2D surface prior to binding to it.

## 3. Results

### 3.1. DNA on Mica Conformational Analysis

In order to address conformational changes of double-stranded DNA upon interaction with the MCM protein complex to distinguish between different binding modes, we initially characterized the conformation of DNA alone on the mica surface by analyzing the end-to-end distance/contour length correlation, as described in Materials and Methods. Representative AFM images taken in air conditions are shown for three different DNA fragments, with lengths of 495 bp, 807 bp, and 1350 bp respectively, deposited on freshly cleaved mica in the presence of Mg^2+^ ions (Figure 2a–c, see Section 2 Materials and Methods) taken in air conditions are shown.

The average values of the mean square end-to-end distance values are plotted as a function of the relative contour length L, as derived from the analysis of 42–50 independent images from three different DNA preparation per each strand length (Figure 2d). The WLC analysis carried out using Equation (1) provides a persistence length value of 45.0 ± 0.3 nm, in agreement with measurements available for DNA restriction fragments of similar length and similar ionic strength [35]. As we can see from Figure 2d, longer strands are associated with a higher variability in the R measurement. This is a typical feature of the WLC model, and is due to two effects: an excluded volume effect, which tends to make the end-to-end distance larger than expected; the fact that longer molecules might take a longer time to equilibrate, contributing to a reduction of R. Interestingly, the horizontal standard deviation measured is small, pointing to an accurate determination of the contour length L. Other binding strategies (not shown here), as the coating of the surface with polymers rich in amine groups (e.g., polyornithine or polylysine) promoted stronger binding, which affected DNA conformations. Instead, from the measurements in buffer with the divalent ions, we could conclude that the nucleic acid molecules were irreversibly adsorbed onto the substrate and that the persistence length measured from AFM images was in good agreement with that of worm-like chain polymers at equilibrium in two dimensions [31], confirming therefore that our conditions were suitable to study the morphological and the conformational changes of DNA when interacting with the protein complex.

### 3.2. MCM Protein Complex–DNA Interaction

After having optimized the deposition and imaging condition for DNA, we then focused on the 807 bp long DNA fragments and proceeded to study DNA–protein interactions using the archaeal MCM protein complex. In Figure 3, we observed DNA–MCM complexes deposited from solution on the mica surface and imaged as before in air, in the non-contact mode. Proteins bound to DNA were distinguishable as protrusions in AFM imaging of about 2.5 nm height with respect to the DNA molecules. The total height of the protein (approximately 4 nm from the mica surface) measured in ambient conditions was an underestimation of the real height value, due to the, although gentle, pressure exerted by the tip in the AC scanning mode. The measured value was in agreement with the formation of protein complexes. These protein complexes are shown to bind at different positions along the DNA strands, as indicated by yellow arrows in Figure 3a. These observations were independent on the concentration of the protein complex. For the case reported in Figure 3, i.e., 2 nM DNA and 2 nM MCM complex, from the analysis of about 1500 DNA molecules (from 8 independent experiments) we calculated that about 75% of the DNA strands were interacting with the MCM complex. Moreover, we never observed more than one MCM complex per strand. This value increased (to one/two complex/strand) when we increased the concentration of MCM complex in solution.

We could classify the mechanism of interaction from the point of view of the topological changes in the DNA. In particular we distinguished between molecular complexes bound to DNA without affecting its conformation (Figure 3c) and complexes where MCM was located at kinks on the DNA strands (Figure 3b); these types of abrupt change in the nucleic acid direction were not observable in the AFM images taken on bare DNA deposited on the mica surface. We therefore assumed that these kinks were actually caused by the protein. In order to discriminate between these two binding modes, we increased the pool of analyzed molecules to about 5000 DNA strands. We found out that around 90% of the MCM complexes bound to DNA were in the first class (“no bending”), while the remaining 10% fell into the second category (“bending”). In order to locate the complex along the DNA strand we performed the location analysis of MCM complexes classified as “bending” and “no bending” using more than one hundred molecules per category. The reference binding point was taken at the middle of the sequence and the histogram profiles is reported in Figure 3d. We observe a preferential MCM complex location close to the DNA edge for the “no bending” case and a random distribution of the positions in the “bending” case.

An obvious interpretation of the features seen by AFM is that the no-bending interactions represent complexes where the DNA threads through the channel, consistent with the canonical mode of binding of ring helicases. More intriguing is the observation that some MCM complexes seem to cause a sharp change in the direction of the nucleic acid chain; this suggests that the DNA may wrap around the hexamer and thus change direction. Although the large majority of the studies of MCM in the presence of nucleic acid visualize ssDNA or dsDNA passing through the central channel, there is indeed some evidence for alternative modes of interaction, with either dsDNA wrapping around the N-terminal domain in an “associated” mode [27], or the strand excluded during unwinding wrapping around the hexamer external surface [36].

Biochemical and structural data suggest that MCM can thread both single-stranded and double-stranded DNA, and is capable of translocation along dsDNA [8,10,12]. The initial melting of the DNA and the transition from a double hexamer bound to dsDNA to a single hexamer bound to only one strand is not completely understood and presumably requires the presence ATP and/or additional replication factors [15,16,17]. As neither a nucleotide nor GINS, Cdc45, or MCM10 were present in our experiment, we expected that MthMCM simply encircles the dsDNA, without melting or unwinding.

It has to be stressed that in the cell the MCM helicase is loaded onto dsDNA through a complex series of steps involving numerous other proteins. In the presence of a relatively long dsDNA strand the hexamer may not easily thread along the dsDNA and may thus favor a looser interaction through the external surface; the difficulties in loading would be consistent with the large number of proteins that remain at the very end of the dsDNA strand.

### 3.3. MCM Mutants–DNA Interaction

In order to confirm the above hypothesis on the two different modes of MCM–DNA binding based on AFM topography imaging of DNA conformational changes, we tested specific MCM mutants. As dsDNA wrapping on the outside surface has been proposed to involve mostly the MCM N-terminal domain and particularly subdomain A [27], we used a mutant of the MthMCM where such a domain has been deleted (ΔsA, [18]). Due to the reduced propensity of the ΔsA mutant to bind DNA, we increased the relative protein–DNA ratio to 10:1 (i.e., 2 nM DNA and 20 nM ΔsA mutant). Out of the nearly 1600 DNA molecules analyzed in this experiment, only 20% of DNA was found to interact with the ΔsA mutant, consistent with a lower affinity for dsDNA than the wild-type [2]. We observed that the few ΔsA proteins that interact with nucleic acid were preferably bound to the end of the DNA fragment (Figure 4a), with only 6% of the proteins interacting with DNA in the “bending” conformation.

To further understand the molecular basis of the different binding modes, we used a mutant where two positively charged residues (R227 and K229), which are located in an N-terminal β-hairpin were mutated to alanine (NBH [14,37]). Within the hexamer the hairpin projects towards the centre of the channel and is involved in DNA binding. Additionally, in this case we used a higher protein–DNA ratio (10:1) to compensate for the reduced affinity of the protein for DNA [14,37]. In the presence of the N-terminal β-hairpin mutations, we analyzed over 1600 DNA molecules, and among these only 18% interacted with the protein. All of the complexes observed corresponded to the “bending” configuration (Figure 4b), where the DNA that seems to be wrapped around the protein complex.

### 3.4. DNA Contour Length Variation

In order to get further details on DNA–protein interaction, we measured the DNA contour length of the bare 807 bp DNA fragment (corresponding to 274.4 nm contour length) and when in interaction with the different protein complexes (Figure 5). In this analysis, we considered only DNA molecules with a single protein bound (more than 95% of the cases for all samples).

We compared the median (from the boxplot in Figure 5a) and the statistical distribution as shown in Figure 5b–e of DNA alone and DNA bound to protein complexes. While DNA alone shows a quasinormal statistical distribution of contour lengths with a median value of 266 ± 7 nm, when interacting with the proteins a higher degree of dispersion and skewness appeared in the data, which justifies a non-parametric analysis. We should point out here that we measured an average “compaction” of DNA with respect to the theoretical value of 274.4 nm that accounts for the difficulty to precisely follow the contour length along the strand in the AFM digital image. For the DNA–MCM and the DNA–ΔsA complexes, we observed a slightly more compact DNA contour length, of 258 ± 9 nm and 255 ± 11 nm respectively, and a longer tail towards shorter lengths. In the case of the DNA–NBH complex we observe a more significant shortening (248 ± 15 nm) and the presence of species that are considerably shorter, consistent with a model where the DNA wraps around the hexamer. In Figure 6 we show possible configurations of DNA wrapping DNA around the protein complex. Given the 13 nm diameter of the protein, we could easily account for such compaction in the “bending” interaction mode, whose exact number will depend on the partial (Figure 6a,b) or complete (Figure 6c) wrapping of DNA around the protein complex. This variability justifies the broader distribution of contour length values reported in the box plot for DNA–NBH. In addition, we cannot discard salt effects, as described before.

### 3.5. DNA Bending Angle Analysis

In addition to the DNA contour length analysis, we performed a statistical analysis of the protein-induced DNA kinks based on the bending angle (β) as shown in Figure 7a. For the DNA–MCM complex, the distribution is centered around 100°, 16° smaller compared to the unbound DNA distribution (Figure 7b). As a comparison, an electron microscopy analysis of MthMCM in complex with a 5.6 kbp double stranded DNA fragment showed a broad distribution of bending angles centered around 90° [27]. Regarding the DNA–ΔsA complex, our data confirm that in terms of bending angle it is behaving as bare DNA (Figure 7c).

For the DNA–NBH complex, the distribution was centered around 97° (Figure 7d), which was not significantly different from the DNA–MCM distribution. Both the MCM complex or NBH mutant were increasing the bending angle of the DNA fragment. On the other end the binding of the ΔsA mutant did not affect the average angle observed for the DNA alone. All these results confirm the visual observation that the interaction with the MCM molecules caused a bend in the DNA.

## 4. Discussion

The main goal of this work was to confirm and support the models of MCM–dsDNA interaction in the near physiological condition, by atomic force microscopy (AFM) single molecule imaging. We studied archaeal MCM from *Methanothermobacter thermautotrophicus* as a simplified model system, interacting with blunt-ended, double-stranded DNA fragments of different lengths. First, we optimized the protocol for AFM imaging to obtain clear and high-resolution images of surface-equilibrated DNA molecules before and after the interaction with a protein complex. Direct visualization of the MCM complexes bound to the DNA, allowed us to distinguish between two different configurations by means of an accurate analysis of AFM topographic images. In one configuration (which we interpreted as the hexamer loaded onto dsDNA) the complex did not change the DNA direction, while in the other (which we interpreted as dsDNA wrapping around the external surface of the hexamer) the binding of the hexamer caused a sharp bend.

This assignment was tested through the study of the separate effects of mutations in the inner channel (NBH mutant) or in the outer ring (ΔsA mutant) of the MCM on the DNA conformation. Our AFM analysis confirmed the preference of the NBH mutant for a wrapping-type interaction and of the ΔsA mutant for a loaded-type interaction. In line with the lower affinity of both mutants for DNA, the total number of DNA bound molecules significantly decreased. From a statistical analysis of the AFM images, we calculated the DNA contour length (L) and bending angle (β) before and after the interaction with the protein complex. We found out that the presence of a single protein complex bound on the DNA reduced the DNA contour length. The reduction was consistent with the DNA wrapping around the protein complex, and was in line with the change of the distribution of the DNA binding angles in the presence of a protein complex.

A variety of possible interactions between MCM and DNA have been reported. The main site of interaction between hexameric helicases and nucleic acid was the central channel, with the nucleic acid threading through the channel (Figure 8a). In the case of MCM there was evidence that the central channel was able to bind both dsDNA and ssDNA: structures of archaeal and eukaryotic MCM complexes bound to both ssDNA [38,39,40,41] and dsDNA [9,13,22,23,42] were obtained by crystallography or electron microscopy.

However, there is also some evidence that MCM is able to bind nucleic acid on the outside surface of the ring, either as an initial association step before the active loading (Figure 8b) [27,43], or during unwinding through a steric exclusion and wrapping (SEW) mechanism (Figure 8c) [36,44,45]. A possible interpretation of the two modes of binding observed by AFM is that in one mode the nucleic acid threads through the ring, so that DNA is actually bound to the central channel in a nearly extended conformation, while in the other mode the DNA remained outside the channel and wraps around the external surface, as seen in CryoEM [27], causing a bend in the DNA (Figure 8b). Although the SEW mechanism proposed the wrapping of a DNA single strand following strand separation, it implies the presence of a nucleic acid binding surface on the external face of the hexamer, along a path that partially wraps around the hexamer; it is possible to conceive that when the dsDNA did not thread inside the channel there may be some association between dsDNA and the external positive surface (Figure 8c).

The analysis of the interaction between protein and nucleic acid in near physiological conditions can be challenging and often requires the contribution of a variety of biochemical and biophysical tools. Atomic force microscopy in particular can provide a quantitative, direct visualization of the interaction, with the capability of controlling the equilibration of DNA molecules on surfaces through careful biomechanical measurements on chemically well defined surfaces, to guarantee the reproducibility of the data. This work provides the basis for further studies on the interactions of MCM and CMG complexes on different DNA substrates (blunt-ended dsDNA, replication forks, 3′ tailed substrates and replication bubbles), in the presence of various nucleotides (ADP/ATP, non-hydrolysable/transition-state analogues) by moving to fast-scan AFM measurements in the physiological, controlled environment.

## Figures and Tables

**Figure 1 materials-14-00687-f001:**
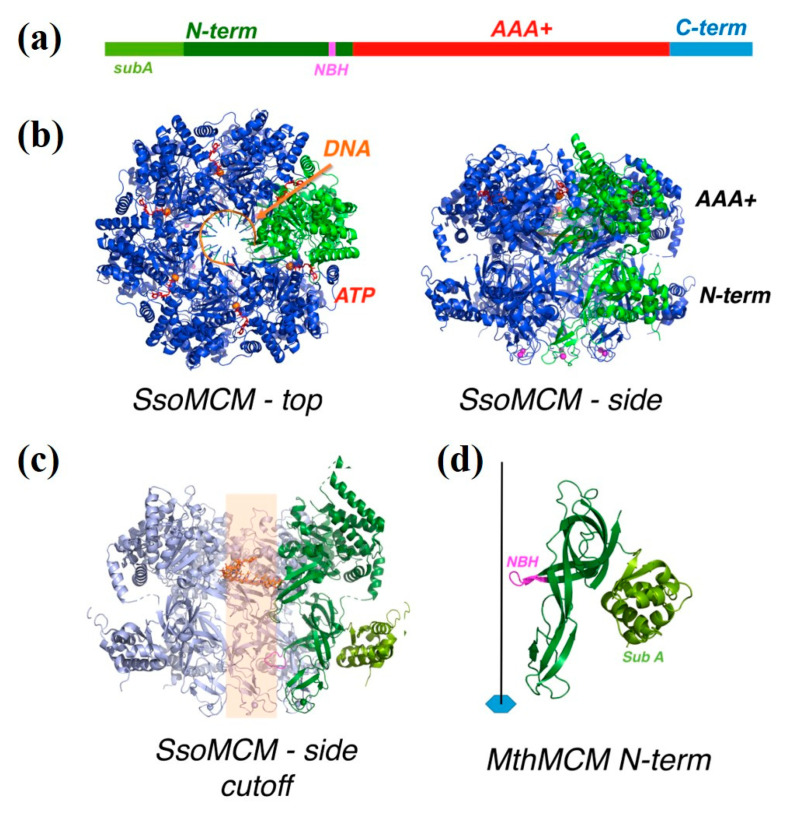
(**a**) A schematic diagram of the architecture of a minichromosome maintenance (MCM) monomer, showing the N-terminal domain in green, the AAA+ domain in red and the C-terminal domain in blue. Within the N-terminal domain, subdomain A and the N-terminal β-hairpin (NBH) are highlighted in light green and magenta, respectively. The C-terminal domain is not visible in the structure. (**b**) The three-dimensional structure of an archaeal MCM hexamer (*Sulfolobus. solfataricus* MCM (SsoMCM), PDB ID: 6MII [13]) bound to DNA, viewed from the top and from the side: one monomer is highlighted in green; the DNA is shown in orange; the ATP bound at the interface between monomers is in red. (**c**) A side view of the SsoMCM hexamer, with the two front monomers removed, so as to see the interior of the channel; a short stretch of DNA is seen bound to the AAA+ domain, but is expected to thread the whole channel (shown by the semitransparent orange stripe); on the green monomer, subdomain A is highlighted in lighter green, and the N-terminal β-hairpin, projecting into the channel, in magenta. (**d**) A close-up of the N-terminal domain from *Methanothermobacter thermoautotrophicus* MCM (PDB ID: 1LTL [14]), with the same color code as panel A and C; the position of the hexameric axis (which coincides with the central channel) is shown.

**Figure 2 materials-14-00687-f002:**
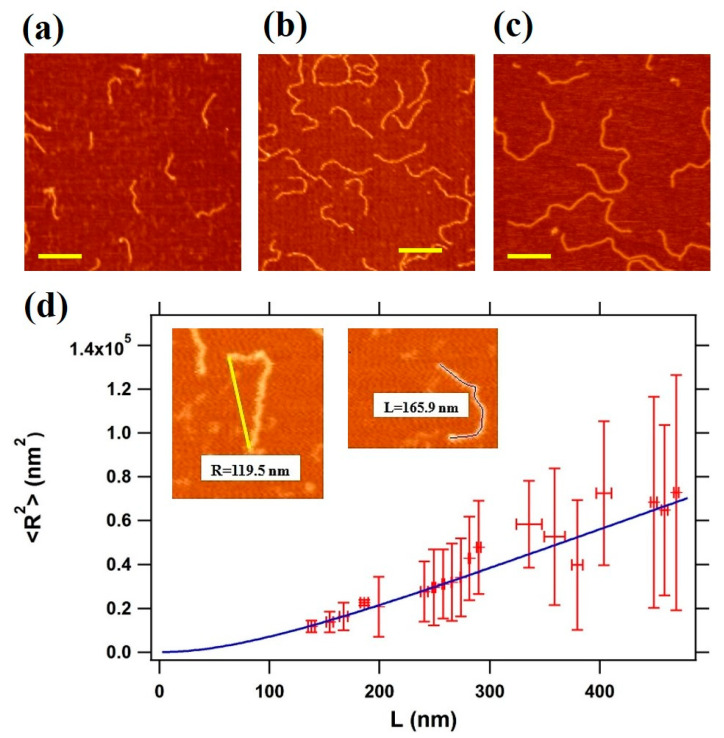
(**a**–**c**) Representative non-contact atomic force microscopy (AFM) images of DNA on mica performed in air. DNA molecules of 2 nM of various lengths (**a**: 495 bp DNA, **b**: 807 bp DNA, and **c**: 1350 bp) in buffer solution were incubated for 1 min at room temperature. The scale bar corresponds to 200 nm; (**d**) the mean square end-to-end distance <R^2^> as a function of the contour length (L), as calculated from AFM 138 images; the blue line is the fitting curve obtained using Equation (1), which gives a persistence length of 45.0 ± 0.3 nm.

**Figure 3 materials-14-00687-f003:**
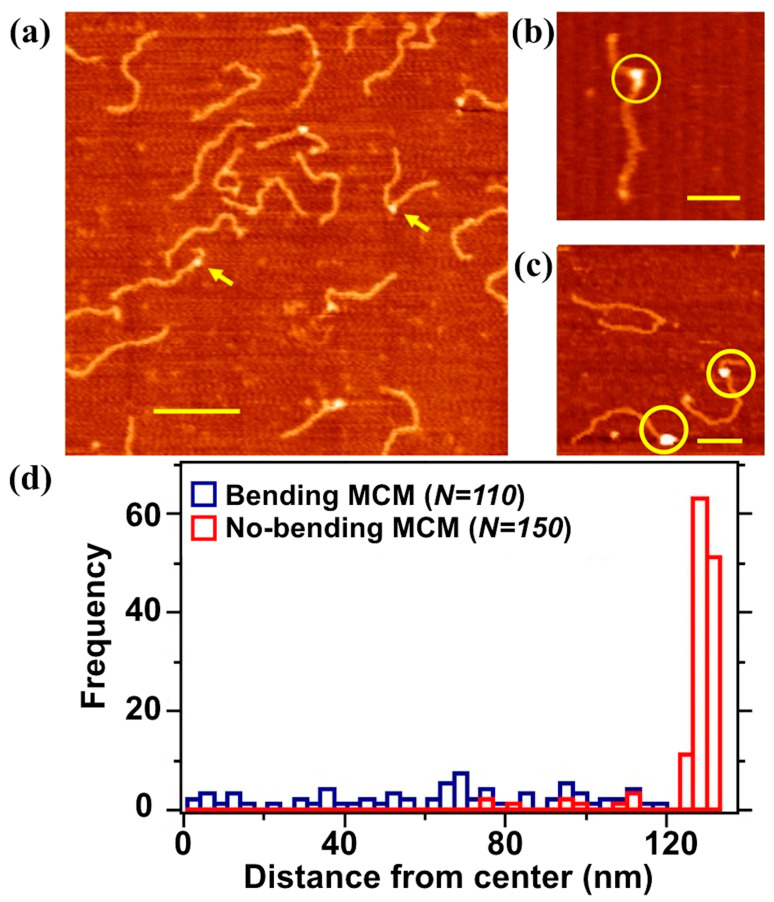
Representative non-contact AFM image of DNA–MCM complexes performed in air. (**a**) A solution containing 2 nM MCM proteins and 2 nM of 807 bp long DNA molecule was deposited on freshly cleaved mica. Scale bar 200 nm. The yellow arrows point to two different MCM–DNA topologies, in which MCM is either bound without changing the direction of the DNA or is creating a kink in the DNA conformation; these kinks are not observed in the AFM images of free DNA. Free DNA molecules that did not interacted with the MCM complex are also visible; (**b**) “bending” MCM: we observe a kink in the DNA induced by the presence of MCM; (**c**) “no bending” MCM: the MCM complex does not affect DNA conformation; and (**d**) distribution of the MCM position along the DNA in the “no bending” and “bending” conformations. MCM position is calculated from the middle of the DNA strand.

**Figure 4 materials-14-00687-f004:**
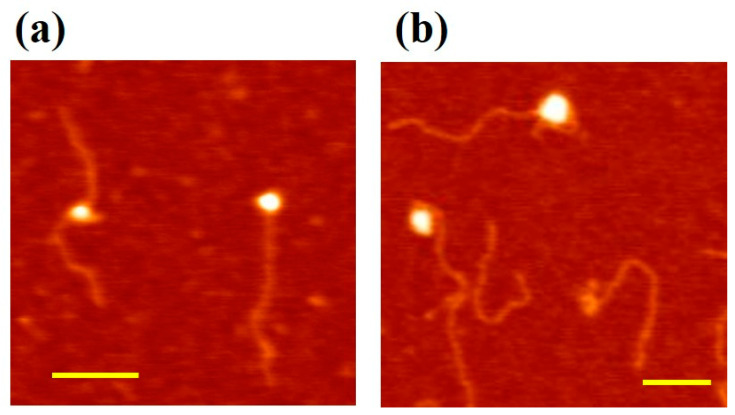
Interaction of MCM mutants with DNA. Here we increased the relative ratio protein/DNA to 10:1 (2 nM 807 bp DNA fragments and 20 nM protein complex). Scale bar 100 nm. (**a**) ∆sA–DNA complex–DNA interaction. The ∆sA complex lacks the N-terminal subdomain A, which has been proposed to be involved in wrapping. From the images, we observed that the protein is preferably bound to the end of the DNA fragment; (**b**) NBH–DNA complex–DNA interaction. The NBH mutant has two mutations in the N-terminal β-hairpins, projecting into the centre of the ring, thus affecting the interaction between DNA and the central channel.

**Figure 5 materials-14-00687-f005:**
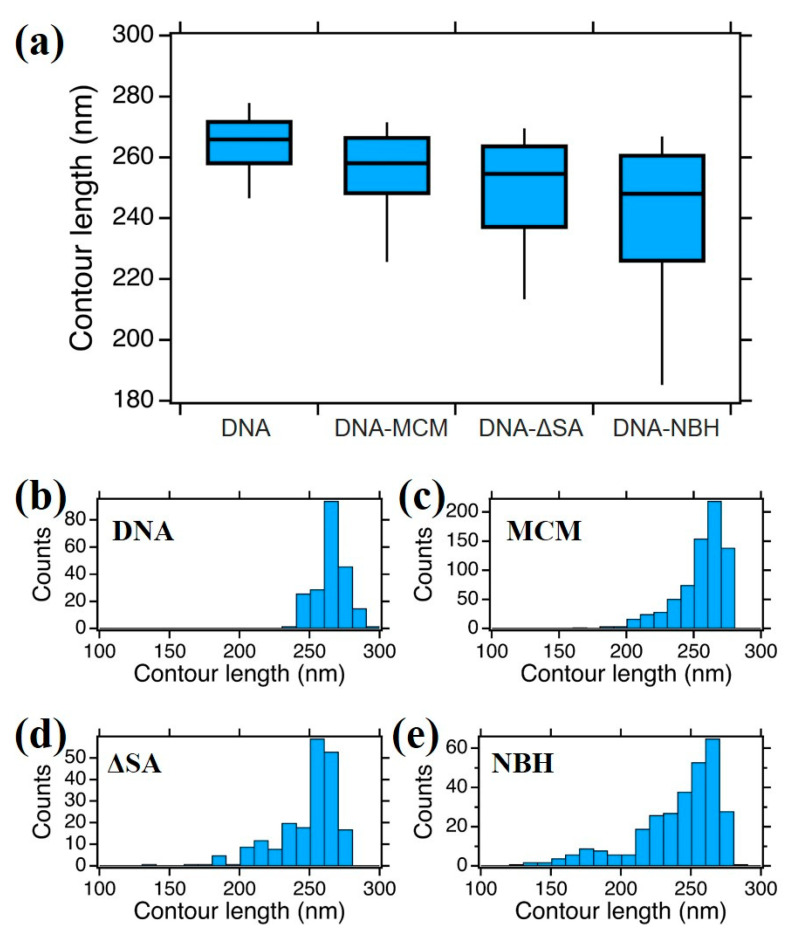
(**a**) Boxplot analysis of DNA contour length of bare DNA, MCM–DNA, ∆sA–DNA, and NBH–DNA complexes; (**b**–**e**) the histograms show the 807 bp DNA contour length distributions of DNA alone (**b**) and protein–DNA complexes (**c**–**e**), constructed with bin size of 10 nm. The distributions are statistically different, with *p* < 0.001 (non parametric Matt–Whitney test).

**Figure 6 materials-14-00687-f006:**
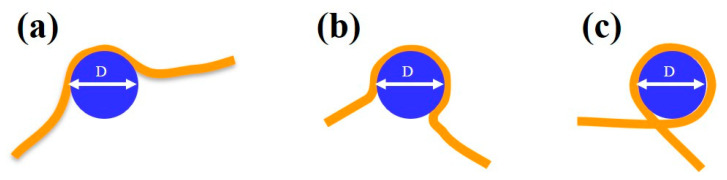
Schematic diagram showing possible modalities in which the DNA (in orange) can wrap around the MCM complex (in blue), accounting for the observed shortened length. (**a,b**) Partial wrapping and (**c**) complete wrapping. The MCM complex has a diameter of 13 nm.

**Figure 7 materials-14-00687-f007:**
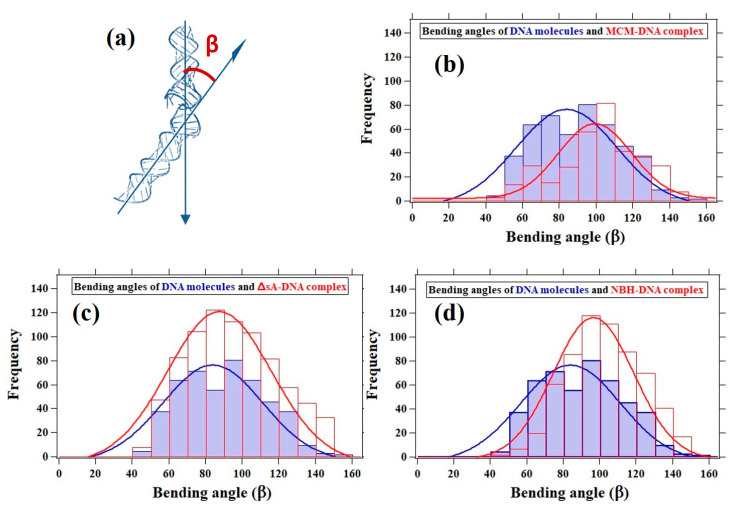
Histograms of the bending angles characterizing the bare DNA (blue distribution), which is 84° and protein-mediated DNA kinks (red distribution). (**a**) The bending angle (β), (**b**) for the MCM–DNA complex, the distribution is centered around 100° shifted by 16° comparing to the bare DNA distribution; (**c**) ∆sA–DNA complex, behaving as bare DNA; and (**d**) NBH–DNA complex, the distribution centered around 97°, which is close to the MCM–DNA distribution.

**Figure 8 materials-14-00687-f008:**
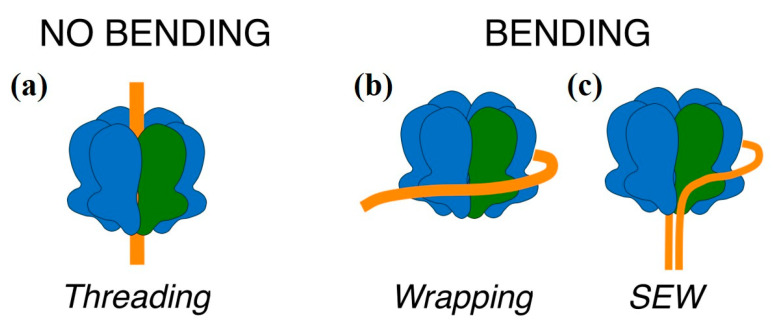
A schematic representation of the possible modes of interaction between MCM helicases and DNA. The MCM helicase is shown in blue, with a single monomer highlighted in green (as in Figure 1) and the DNA in orange. (**a**) The canonical mode of binding of hexameric ring helicases/translocases, with the ssDNA or dsDNA threading through the central channel; (**b**) the dsDNA wrapping around the external surface of the ring and interacting with subdomain A, as visualized by a low resolution CryoEM analysis [26]; (**c**) the strand-exclusion and wrapping model (SEW) in which one strand threads through the central channel, and the excluded strand wraps around the external surface of the hexamer.

## Data Availability

The data presented in this study are available upon request from the corresponding authors.

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
