# Peer review of "Atomic Force Microscopy Investigation of the Interactions between the MCM Helicase and DNA"

_materials, 2021, doi:10.3390/ma14030687_

Round 1

Reviewer 1 Report

Review of “Atomic Force Microscopy investigation of the interactions between the MCM helicase and DNA” by Khalid et al for Materials. This manuscript utilizes AFM to measure the contour length and bending of double stranded DNA after incubation with the MthMCM helicase. These hexameric helicases are essential for binding to and unwinding DNA at the front of the replication fork. Their interaction with DNA is still not completely defined. Even though, most would agree that the MCM helicase must encircle and translocate on 1 strand to unwind DNA, how interacts with duplex DNA or transitions from duplex encircling to single-strand encircling is not known. The authors provide some additional evidence for interactions and bending of DNA when MCM is bound. The manuscript is solid and I have only a few clarifying comments that need to be addressed.

Major Comments:

  • Can you confirm that the MCM proteins are hexamer when bound to DNA by AFM, or could they be monomer or dimer or something else? Maybe this can be done by extracting the height profile compared to some standard proteins?
  • Page 7, line 231 to 232. And again lines 245-246. Can you quantify the MCM location on DNA using a histogram profile to show this more directly? And I also imagine/wonder whether those molecules that are kinking the DNA (exterior binding) are more distributed in the middle. Is this random or significant? This should be shown more directly in a plot of the data with significance determined.

Minor Comments:

  • Page 12, line 368, should this be “Exclusion”

Reviewer 2 Report

Review of Materials #1062207

The Manuscript by Khalid, et al established initial conditions to study MCM helicase and DNA interactions by AFM. Building on other single molecule techniques and structural analyses, AFM is a complementary tool to understand protein (MCM) and DNA interactions. The authors optimized AFM surface conditions and then observed direct MCM-DNA interactions and variations in bending. Using MCM binding mutants they further tested binding interactions. They interpreted the sharp bends to be external wrapping of DNA around the outside of MCM. In general, the paper was well written for a general audience not just AFM experts. I would support publication as a proof-of-concept paper that will contribute to the field of nucleic acid replication enzymes and lays the foundation for further studies. For revision, I would suggest addressing the following minor issues.

1. Figure 6. The colors in the Legend do not match the figure illustration.

2. Please discuss how these results might inform biological relevance:

a.  In particular, the MCM first has to encircle dsDNA and then somehow melts dsDNA to encircle ssDNA for processive unwinding. Do the observed structures help understand that transition from dsDNA to ssDNA?

b. Does the sharp bending due to external nucleic acid binding sites occur in the cell? Does that conformation have a biological function or is it an artifact of the in vitro method?

3. Is the bending angle consistent with the DNA bend observed in Ali, 2017 as measured by cryoEM? (Abid Ali, F., Douglas, M.E., Locke, J. et al. Cryo-EM structure of a licensed DNA replication origin. Nat Commun 8, 2241 (2017). https://doi.org/10.1038/s41467-017-02389-0)

4. For future experiments, you mentioned other DNA replication protein complexes you plan to test but what DNA substrates would also be relevant? i.e. a linear DNA with a replication fork on one end or a substrate with a replication bubble?
